# The role of perceived threat and self-efficacy in the use of Insecticide Treated Bednets (ITNs) to prevent malaria among pregnant women in Tororo District, Uganda

**Charles Nelson Kakaire**📷*, **Nicola Christofides**

School of Public Health, Faculty of Health Sciences, University of Witwatersrand, Johanesburg, South Africa

* ckakaire@gmail.com

## Abstract

### Background

Despite increased coverage of Insecticide Treated Nets (ITNs) due to free distribution programs, ITN use in Uganda remains sub optimal among pregnant women. This study explored the relationship between constructs of a theoretical framework and Net use.

### Objective

The study examined the role of constructs from the Extended Parallel Process Model (EPPM) in determining ITN use amongst pregnant women 15–49 years in Tororo district, Uganda.

### Methods

A cross-sectional study using a systematic sample was conducted among 230 pregnant women attending antenatal care. The questionnaire was administered by trained research assistants. Analysis was conducted to establish the relationship between ITN use and perceived susceptibility, severity, self-efficacy and response efficacy.

### Results

Over three-quarters (78.6%) reported using ITNs the night before the study while 49.78% reported consistent Net use. High self-efficacy (AOR 9.48 95%CI 3.34–26.91) was associated with ITN use the previous night and consistent use. High perceived threat was associated with consistent ITN use (AOR 2.78, 95%CI 1.16–6.67) but not with Net use the previous night.

### Conclusion

Self-efficacy was an important predictor of ITN use, as well as high levels of fear, as measured through perceived threat, which was associated with consistent ITN use, but not ITN

**Data Availability Statement:** All relevant data are within the paper and its Supporting Information files.

**Funding:** While the study received a partial funding contribution from the Malaria consortium Uganda equivalent to approx. $2300, the funders had no role in study design, data collection and analysis, decision to publish, or preparation of the manuscript.

**Competing interests:** The authors have declared that no competing interests exist.

use the previous night. Social and behavior change communication interventions should focus on improving self-efficacy to use ITNs.

## Introduction

Malaria is a public health concern in sub-Saharan Africa, accounting for approximately 93% of the global estimated 228 million cases in 2018 [1]. Uganda has one of the highest malaria transmission rates in the region and contributes significantly to the global burden [2]. The disease is responsible for 30–50% of outpatient cases, 15–20% of inpatient admissions to hospitals, and one in every five reported deaths, with pregnant women and children under the age of five at greatest risk [3]. Malaria is endemic across Uganda with the only exception being the southwestern part of the country. While malaria prevalence in the country has reduced over the last two decades, Uganda remains one of the high-burden countries with malaria incidence rates at above 154 cases per 1,000 people in nearly all districts [4].

Pregnant women are at high risk of malaria transmission. Several outcomes are associated with malaria in pregnancy including stillbirth, preterm birth, maternal and neonatal mortality, congenital malaria, maternal anaemia and low birth weight [5]. A study by Namusoke and others noted that malaria infection rates among pregnant women were 15.5% (59/380) of active infections and 4.5% (17/380) of past infections [6]. The 2018–19 Malaria indicator survey revealed that 65% of pregnant women slept under an ITN the previous night. The two main strategies to prevent malaria during pregnancy are Intermittent Preventive Treatment of malaria in pregnancy (IPTp) and use of Insecticide Treated mosquito Nets (ITNs) [3]. Insecticide Treated mosquito nets are recommended by the World Health Organization (WHO) because of their effectiveness. The evidence for efficacy of ITNs in reducing cases of malaria is widely documented [7, 8]. There are different strategies for distribution of ITNs in Uganda, but the main ones are free distribution to vulnerable groups (including children under five years and pregnant women) through antenatal care (ANC) clinics, and as a result ITN ownership in endemic areas is high [9]. While the WHO notes that global ITN coverage and use has increased dramatically; there are variations at the country level, especially in Africa [10]. Cognitive, social and emotional factors influencing bahavior have an important role to play in determining consistent ITN use especially among high risk groups such as pregnant women [11].

Theoretical frameworks provide a basis for determining factors that may predict behavioral outcomes. The Extended Parallel Process Model (EPPM) is premised on using fear appeals to influence individuals to adopt health behaviors [12, 13]. The model posits that fear appeals are most effective when they increase people's perception of the threat of the disease and at the same time increase efficacy to use the preventive strategy [14].

The EPPM postulates that a combination of threat, defined as an individual's perception of both severity (beliefs about the magnitude of a threat) and susceptibility (beliefs about one's risk of experiencing the threat), and efficacy—composed of an individual's perception of both response efficacy (beliefs about the effectiveness of a recommended action in averting threat) and self-efficacy (beliefs about one's ability to perform the recommended response to avert the threat)—determines behavior change (Fig 1) [13]. While the EPPM has been tested in several interventions including workplace safety, asthma intervention programs, and physicians' decision to test patients for kidney disease and more recently COVID-19 vaccination uptake [15–18]. While there is no known study of the application to ITN use among pregnant women,

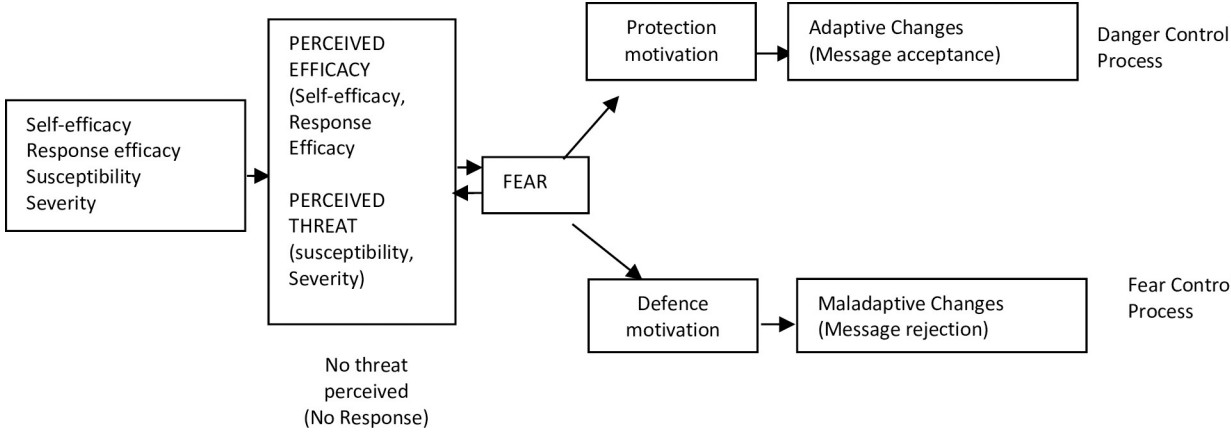

**Fig 1. The extended parallel process model.**

EPPM has been used to guide malaria prevention interventions for school children and care-givers in Ghana [19, 20]. The EPPM predicts that fear of a health risk (such as malaria) can cause either adaptive, self-protective actions leading to behavior change (ITN use in this case) or maladaptive, self-defeating actions [13].

For interventions to be effective, communication messages should be aimed at increasing risk perceptions related to the threat of malaria, specifically perceptions that the disease can have severe outcomes and that everyone is susceptible, with pregnant women and children under the age of five being particularly vulnerable. This should be accompanied with appropriate efficacy messages to ensure danger control, including self-efficacy to use ITNs and promoting perceptions that ITNs are effective in controlling the transmission of malaria. Combining the four EPPM constructs is theoretically a recipe for behavioral change. In their study, Roberto et al. confirmed their prediction that physicians who perceived greater threat to patients and greater efficacy demonstrated greater intentions and behavior to test their patients' level of kidney functioning [15].

This study explored the relationship between the four EPPM constructs and their association with ITN use among pregnant women in the Tororo district in Uganda.

## Methods

### Study area and population

A cross-sectional study was conducted in November 2015. Data were collected from 230 pregnant women attending antenatal services within the Tororo district general hospital in Eastern Uganda, approximately 230km from the capital, Kampala (see Fig 2). Tororo district has an exceptionally high burden of malaria [21]. Although the district outpatient attendance rates related to malaria are similar to the national average, its entomological inoculation rate (EIR) is one of the highest in Uganda, estimated at 562 effective bites per person per year—an average exposure of 1.5 infectious mosquito bites per night [22].

The antenatal care (ANC) clinic at Tororo general hospital distributes free ITNs to pregnant women from the Malaria Consortium, a non-profit organization specializing in the prevention, control and treatment of malaria and other communicable diseases among vulnerable populations. A sample size of 212 was estimated based on power of 0.80 and alpha of 0.05. It assumed that 35% of women with low perceived severity would use ITNs and 55% among those with high perceived severity and efficacy. This was based on ITN use among pregnant

## Map of Uganda showing Tororo District

**Fig 2. Map of Uganda showing Tororo district–the study area.**

women at the time. We added 20% to account for non-response which resulted in a final sample size of 233. Systematic sampling was utilized to select study participants in the health care facilities. Every third woman in the queue for antenatal care was approached and invited to participate, and face-to-face interviews were conducted by trained female research assistants

daily over a period of three weeks. All women who were able to understand and speak Japadhola, the local language, were eligible for inclusion. Data were collected using a questionnaire adapted from the Malaria Behavior Change Communication (BCC) Indicator reference guide [21]. The questionnaire was translated into Japadhola—the main local language spoken in Tororo district and pretested prior to data collection. The Japadhola version was then translated back into English to determine accuracy. Informed written consent was sought from each participant prior to the interview.

## Questionnaire

The questionnaire measured ITN use as the outcome of interest, using two dependent variables: use of an ITN the previous night and consistent ITN use, which included participants who reported that they always use an ITN. All other responses (mostly, sometimes and never) were classified as inconsistent use.

The ideational factors based on the EPPM constructs included perceived susceptibility, perceived severity, perceived threat, self-efficacy, and response efficacy.

Perceived susceptibility was measured by asking participants the extent to which they felt at risk of acquiring malaria using six items with response categories on a 4-point Likert scale with answers ranging from 1 (*strongly agree*) to 4 (*strongly disagree*), *e.g. People only get malaria when there are lots of mosquitoes* (Cronbach's Alpha = 0.71). Perceived severity was measured with a single item on a 4-point scale ranging from 1 (*strongly agree*) to 4 (*strongly disagree*), *e.g. Every case of malaria can potentially lead to death*. The score was categorised into tertiles representing high, moderate and low perceived susceptibility. Perceived threat was a variable that combined the perceived susceptibility sub-scale and an item measuring perceived severity (Cronbach's Alpha = 0.68).

Self-efficacy was measured using four items with response categories on a 4-point scale ranging from 1 (*definitely could*) to 4 (*definitely could not*), e.g. *I am confident that I can sleep under a bed net for the entire night even when there are few mosquitoes* (Cronbach's Alpha = 0.75). The score was categorised into tertiles representing high, moderate and low self-efficacy.

Response efficacy was measured by asking participants about their beliefs that ITNs can prevent getting malaria with responses recorded from four items using a 4-point scale ranging from 1 (*strongly agree*) to 4 (*strongly disagree*), e.g. *Bed nets only prevent mosquito bites when used with certain types of beds* (Cronbach's Alpha = 0.56). The score was categorised into tertiles representing high, moderate and low response efficacy. The two sub-scales were combined to create an efficacy scale (Cronbach Alpha = 0.71).

Covariates including socio demographic characteristics—age, education, marital status, and socio-economic status—were measured. Educational attainment was measured as the highest level of schooling completed (including an option of none), and marital status was measured through a single item (currently married or living together with someone as if married). Socio-economic status was measured with an asset-based index, used to calculate household welfare in surveys in low and middle-income countries. Data was collected on availability of electricity in the household, and the ownership of items such as a radio, mobile phone and refrigerator, assets that are believed to have a strong association with poverty levels [16]. A score for socioeconomic status was generated and transformed into categorical data presented as low, medium and high.

Ethical approval was obtained from the University of Witwatersrand Faculty of Health Sciences Human Research Ethics Committee, in South Africa (#M140860), the School of Health Sciences Research and Ethics committee at Makerere University in Uganda (#256) and the Uganda National Council for Science and Technology (# SS 3656).

## Statistical analysis

Data were analyzed using STATA version 15 (STATA Corporation, College Station, Texas USA). Frequency tables were used to present descriptive data on socio-demographic characteristics, ITN use and the independent variables of threat and efficacy while inferential statistics were used to compare ITN users and non-users' perceived susceptibility, perceived severity, self-efficacy and response efficacy, and were scored and transformed into categorical variables of low, medium and high based on tertiles. Pearson's chi-square or Fisher's exact tests were used to detect a relationship between socio-demographic characteristics of the study population and their reported use of ITNs and consistent ITN use. Multivariable, binary logistic regression models were then built of the two outcomes (ITN use the previous night and consistent use) to further test the associations between the key theoretical constructs while controlling for possible confounders such as socio-economic status and educational attainment.

# Results

## Socio demographic characteristics of pregnant women

A total of two hundred thirty pregnant women (n = 230) participated in the study. Table 1 shows the mean age of the study participants was 25 years with a range of 15 to 49 years. Nearly all (94.81%) had some schooling with 36.32% having reached secondary school. A third

**Table 1. Socio demographic characteristics.**

| Characteristic | Frequency (n) | Percentage (%) |
|---|---|---|
| **Age (n = 230)** | | |
| 15–24 | 111 | 48.3 |
| 25–34 | 92 | 40.0 |
| >34 | 27 | 11.7 |
| *Mean years (range)* | *25 (15–49)* | |
| **Educational attainment (n = 202)** | | |
| No school | 11 | 5.5 |
| Primary only | 85 | 42.1 |
| Secondary incomplete | 74 | 36.6 |
| Completed high school or tertiary | 32 | 15.8 |
| Missing data | 28 | |
| **Socio Economic Status (n = 230)** | | |
| Low | 53 | 23.1 |
| Medium | 130 | 56.5 |
| High | 47 | 20.4 |
| **Currently married or living together as if married (n = 230)** | | |
| Yes | *220* | *95.6* |
| No | *10* | *4.4* |
| **Residence (n = 230)** | | |
| Urban | 90 | 39.1 |
| Rural | 140 | 60.9 |
| **Number of children in home (n = 230)** | | |
| 0–2 | 126 | 54.8 |
| 3–5 | 74 | 32.2 |
| >5 | 30 | 13.0 |
| *Mean number of children (sd)* | *2.92(1.94)* | |

(33.49%) had incomplete primary education and 10.38% had studied beyond secondary school.

While all the participants came from poorer households (as per the scale categorisation), there was some variation. Most of the pregnant women (56.52%) were found to be in the middle socio-economic status category, followed by 23.04% in the lower and 20.43% in the high category. Nearly all (95.65%) were married or living together with a partner.

Most (60.87%) were from rural areas with only 39.13% residing in urban locations. More than half (54.78%) were pregnant with their first child or had one or two other children, 32.17% had three to five children, and 13.04% (30) more than five children.

### ITN use among pregnant women

Most of the pregnant women (78.6%; n = 180) reported sleeping under an ITN the night before the study. Nearly half (49.8%) reported using the nets every night compared to 23.58% most nights, 19.21% who reported using the ITNs some nights and the remaining 7.42% reported never sleeping in an ITN (Table 2).

Among the 49 participants who did not sleep under an ITN the night before the study, 46 provided a reason. The common reasons given included net being too hot (23.91%), worn-out or in poor condition (21.74%), not enough nets in the house (10.87%), net not hung (6.62%) and net being too cold or used by someone else (both at 2.17%). Nearly a third of the non-net users (32.61%) mentioned other reasons for non-use such as visiting a friend where there was no net and spending the night at a funeral.

### Relationship between socio demographic characteristics, threat, efficacy and ITN use among pregnant women

Among the socio-demographic characteristics, only socio-economic status (SES) was significantly related to both use of ITNs the previous night (p = 0.04) and consistent ITN use

**Table 2. Insecticide treated net use among pregnant women.**

| ITN use | Frequency(n) | Percentage (%) |
|---|---|---|
| **Slept Under ITN last night** *(n = 229)* | | |
| Yes | 180 | 78.6 |
| No | 49 | 21.4 |
| **Consistent ITN use** | | |
| Yes | 114 | 49.8 |
| No | 115 | 50.2 |
| **How often sleep in ITN** *(n = 229)* | | |
| Every night | 114 | 49.8 |
| Most nights | 54 | 23.6 |
| Some nights | 44 | 19.2 |
| Never | 17 | 7.4 |
| **Reasons for non ITN use** *(n = 46)* | | |
| Too Hot | 11 | 23.9 |
| Too cold | 1 | 2.2 |
| Not enough ITNs | 5 | 10.9 |
| ITN not hung up | 3 | 6.5 |
| ITN used by someone else | 1 | 2.2 |
| ITN worn out/poor condition | 10 | 21.7 |
| Other (away from home, funerals etc) | 15 | 32.6 |

**Table 3. Relation between socio-demographic characteristics and using an ITN the previous night and consistent ITN use.**

| Characteristic | Used ITN the previous night n (%) | No ITN use the previous night n (%) | p value | Consistent ITN use n (%) | Inconsistent ITN use n (%) | p-value |
|---|---|---|---|---|---|---|
| **Age (n = 229)** | | | | | | |
| 15–24 | 90(81.1) | 21(18.9) | 0.67 | 61(54.9) | 50(45.1) | 0.12 |
| 25–34 | 70(76.1) | 22(23.9) | | 44(48.3) | 47(51.7) | |
| >34 | 20(76.9) | 6(23.1) | | 9(63.3) | 18(66.7) | |
| **Attended formal education (n = 229)** | | | | | | |
| No | 6(54.6) | 5(45.4) | 0.06 | 4(36.4) | 7(63.6) | 0.37 |
| Yes | 174(79.8) | 44(20.2) | | 110(50.5) | 108(49.5) | |
| **Educational level (n = 202)** | | | | | | |
| No formal schooling | 6 (54.6) | 5 (45.4) | | 4(36.4) | 7(63.6) | |
| School Incomplete | 88(80.0) | 24(20.0) | 0.12 | 54(48.2) | 58(51.8) | 0.52 |
| Completed high school or high | 86(81.1) | 20(18.9) | | 56(52.8) | 50(47.2) | |
| **Socio Economic Status (n = 229)** | | | | | | |
| Low | 33(62.3) | 20(37.7) | 0.004 | 16(30.8) | 36(69.2) | |
| Moderate | 107(82.9) | 22(17.1) | | 65(50.0) | 65(50.0) | <0.001 |
| High | 40(85.1) | 7(14.9) | | 33(70.2) | 14(29.8) | |
| **Currently married or living together as if married (n = 229)** | | | | | | |
| Yes | 172(78.5) | 47(21.5) | 0.91 | 6(60.0) | 4(40.0) | 0.51 |
| No | 8(80.0) | 2(20.0) | | 108(49.3) | 111(50.7) | |
| **Residence (n = 229)** | | | | | | |
| Urban | 71(78.9) | 19(21.1) | 0.93 | 50(55.6) | 40(44.4) | 0.16 |
| Rural | 109(78.4) | 30(21.6) | | 75(54.0) | 64(46.0) | |
| **Number of children in home (n = 229)** | | | | | | |
| 0–2 | 100(79.4) | 26(20.6) | 0.92 | 66(52.4) | 60(47.6) | 0.47 |
| 3–5 | 58(78.4) | 16(21.6) | | 36(49.3) | 37(50.7) | |
| >5 | 22(75.9) | 7(24.1) | | 12(40.0) | 18(60.0) | |

(Table 3). Pregnant women in the high SES category were more likely (85.1%) to use ITNs than their counterparts in the moderate (82.9%) and the lower SES category (62.3%) respectively. A similar pattern emerged for consistent ITN use where a high proportion of women with relatively high SES used nets more consistently. Any formal education was marginally associated with ITN use the previous night (p = 0.06) with 79.6% of women using ITNs.

All the other socio demographic characteristics including age, place of residence, number of children, and marital status were not significantly associated with ITN use.

Self-efficacy was related to ITN use the previous night while both perceived threat and perceived efficacy were related to consistent ITN use (Table 4). Pregnant women (n = 185) who perceived malaria to be a greater threat to them and their families were more likely to report consistently using an ITN, 40.2%% compared to 25.5% who reported inconsistent use.

Pregnant women with high self-efficacy were significantly more likely to have used ITNs (54.9%) the previous night compared to those who had not slept under a bednet (20.9%). Similarly, 69.4% of women with high self-efficacy reported using ITNs consistently compared to 25.9% who reported inconsistent use. Efficacy (a combined score of self-efficacy and response efficacy) was significantly associated with ITN use, with the majority (87.10%) with high efficacy reporting ITN use the previous night. Response efficacy by itself was not significantly associated with ITN use.

**Table 4. Relationship between using an ITN the previous night, consistent ITN use and the EPPM constructs.**

| | Used ITN previous night | Did not use ITN previous night | P-value | Consistent ITN use | Inconsistent / no ITN use | P-value |
|---|---|---|---|---|---|---|
| | n (%) | n (%) | | n (%) | n (%) | |
| **Perceived threat (n = 185)** | | | | | | |
| Low | 65 (44.5) | 16(41.0) | 0.92 | 26(29.9) | 55(56.1) | 0.002 |
| Moderate | 34(23.3) | 10(25.6) | | 26(29.9) | 18(18.4) | |
| High | 47(32.2) | 13(33.3) | | 35(40.2) | 25(25.5) | |
| **Self-Efficacy (n = 216)** | | | | | | |
| Low | 23(13.3) | 22(51.3) | <0.001 | 3(2.8) | 42(38.9) | <0.001 |
| Moderate | 55(31.8) | 12(27.9) | | 30(27.8) | 38(35.2) | |
| High | 95(54.9) | 9 (20.9) | | 75(69.4) | 28(25.9) | |
| **Response Efficacy (n = 207)** | | | | | | |
| Low | 26(15.2) | 8 (21.6.51) | 0.37 | 13 (12.2 | 21(20.8) | 0.86 |
| Moderate | 50(29.2) | 13(35.1) | | 38 (20.0) | 34 (33.7) | |
| High | 95(55.6) | 16(43.2) | | 64 (59.8) | 46 (45.5) | |
| **Efficacy (self-efficacy + response efficacy)—n = 203** | | | | | | |
| Low | 42(25.2) | 18(50.0) | 0.01 | 13(12.6) | 48(48.0) | <0.01 |
| Moderate | 50(29.9) | 11(30.6) | | 32(31.1) | 29(29.0) | |
| High | 75(44.9) | 7(19.4) | | 58(56.3) | 23(3.0) | |

Multivariate logistic regression analysis was conducted to test the relationship between ITN use the previous night and threat and efficacy, adjusting for socio-economic status and education. The results of the model (n = 229) are presented in Table 5.

After adjusting for other covariates, self-efficacy was significantly associated with ITN use the previous night. Pregnant women with high self-efficacy were nine times more likely to use ITNs than those with low self-efficacy (AOR 9.48). High socioeconomic status was also associated with consistent ITN use. All analyses controlled for formal education and socio-economic status as potential confounders.

**Table 5. Logistic regression model for using an ITN the previous night and EPPM constructs, adjusting for age and education and socio-economic status (n = 176).**

| Variable | AOR | 95% Confidence Interval | p value |
|---|---|---|---|
| Self-efficacy | 9.48 | 3.34–26.91 | <0.001 |
| Response efficacy | 1.73 | 0.71–4.24 | 0.23 |
| Threat Low (Ref.) | - | - | - |
| Moderate | 0.29 | 0.09–0.90 | 0.03 |
| High | 0.49 | 0.15–1.57 | 0.23 |
| **Socio-economic status** | | | |
| • Lower (ref.) | - | - | - |
| • Moderate | 3.24 | 1.17–8.99 | 0.02 |
| • High | 2.06 | 0.61–6.93 | 0.24 |
| **Educational attainment** | | | |
| No school (ref.) | - | - | - |
| Primary | 0.85 | 0.13–5.32 | 0.86 |
| High School | 1.27 | 0.20–8.10 | 0.80 |
| **Age** | | | |
| 15–24 (ref.) | - | | |
| 25–34 | 0.76 | 0.29, 1.98 | 0.07 |
| >34 | 0.75 | 0.19, 2.98 | 0.46 |

**Table 6. Association between EPPM and consistent ITN use controlling for educational attainment, age and SES.**

|  | AOR | 95% Confidence Interval | | P value |
|---|---|---|---|---|
| **Self-efficacy** | 17.03 | 4.57 | 63.42 | 0.00 |
| Perceived threat |  |  |  |  |
| • Low (ref.) | - | - | - |  |
| • Moderate | 2.43 | 0.97 | 6.05 | 0.06 |
| • High | 2.78 | 1.16 | 6.67 | 0.02 |
| **Response-efficacy** | 1.95 | 0.94 | 4.05 | 0.07 |
| **Socio-economic status** |  |  |  |  |
| • Low (ref.) | - | - | - |  |
| • Medium | 3.18 | 1.19 | 8.49 | 0.02 |
| • High | 7.41 | 2.32 | 23.68 | <0.01 |
| **Age** |  |  |  |  |
| • 15–24 (ref.) | - | - | - |  |
| • 25–34 | 0.85 | 0.39 | 1.93 | 0.68 |
| • >34 | 0.43 | 0.13 | 1.93 | 0.17 |
| **Educational attainment** |  |  |  |  |
| • No school (ref.) | - | - | - |  |
| • Primary | 0.26 | 0.03 | 1.93 | 0.19 |
| • High School | 0.36 | 0.05 | 2.17 | 0.32 |

Table 6 shows that consistent ITN use was associated with high self-efficacy. Pregnant women with high self-efficacy were much more likely to use ITNs consistently. Pregnant women with high perceived threat were nearly three times more likely to report consistent ITN use than their counterparts. High socioeconomic status was also associated with consistent ITN use.

## Discussion

This study sought to investigate the relationship between threat and efficacy and use of ITNs among pregnant women. ITN use the night before the survey among the study population was found to be high (78.6%), similar to the 2018/9 Uganda Malaria Indicator survey results which found that 78% of pregnant women 15–49 years of age had slept under an ITN the previous night [23]. Consistent ITN use, reported as "Always using the net", was found to be lower as reported by about half of the participants.

Consistent with theoretical predictions self-efficacy in this study was positively associated with ITN use [12, 24]. Pregnant women with high self-efficacy were more likely to use ITNs than their counterparts. Earlier studies examining the threat and efficacy effect found that self-efficacy was a predictor in the adoption of preventive practices in response to various health communications [25, 26].

Perceived self-efficacy is recognised as an important construct in the adoption and sustained practice of behaviors. Bandura stated that "self-efficacy is the foundation of human action and motivation" [27]. In a net-use study in Tanzania, households with low efficacy were found to have decreased ITN use as their perceived threat of malaria increased. Use of ITNs among those with high efficacy however increased with an increase in threat perceptions, although only to a certain point [28].

Self-efficacy was found to be high among most of the participants and strongly associated with both ITN use the previous night and consistent use. Consistent ITN use was also associated with high perceived threat. This finding was similar to Cho & Witte's Ethiopia study

which found that for bahavior change to occur, both levels of threat and efficacy ought to be high, but that efficacy variables should be high than threat variables [24].

There was a contradiction in the findings between the association of perceived threat (susceptibility and severity) and ITN use the previous night versus consistent ITN use. Although most participants had high perceived susceptibility to malaria, about a third did not perceive themselves to be very susceptible. It is possible that ITN use the previous night was over-reported which would explain the inconsistency. Social and bahavior change (SBC) interventions in malaria endemic areas therefore should consistently aim to increase individual/community risk perception while maintaining high levels of self-efficacy to use ITNs.

High perceived threat (combined severity and susceptibility) was associated with consistent ITN use. This finding indicates a possible danger control response where study participants perceived the consequences of malaria to be severe, and because their self-efficacy was high, took necessary actions to prevent it (by sleeping under an ITN).

Studies have reported that increase perceived severity is associated with the desired behavior [29]. In Vanuatu, high perceptions of malaria severity were linked to sustained use of ITNs on Aneityum Island [30]. Similarly, a risk perception framework to understand health behavior by Rimal & Real posited that the high severity/risk group could experience a high level of anxiety and attempt to reduce it by seeking additional information, leading them to enact healthier behaviours [31]. The role of threat in ITN use was mixed in this study; moderate levels of threats were associated with ITN use the previous night, while high levels of threat were associated with consistent use. Fear appeal authors have argued that ensuring that a threat is regarded as personally relevant by members of the target audience should be the main moderating factor in behavior intention [32].

Perceptions of response efficacy were low among the majority of pregnant women in the study, and this construct on its own was not associated with ITN use the previous night or with consistent ITN use. In studies that investigated behaviors related to other health issues such as HIV prevention, smoking cessation, alcohol reduction and cancer prevention, people were less likely to engage in a specific recommended health bahavior if they did not believe in its effectiveness in solving the problem or achieving a desired outcome [33]. However, a meta-analysis of 65 studies examining the role of high response efficacy found that the construct was linked to behavioural outcomes and intentions [33].

The finding related to low response efficacy could be a result of most malaria prevention programs in the district concentrating on messaging around the threat of the disease rather than the effectiveness of ITNs in reducing malaria transmission. As a result, it is possible that pregnant women's perceptions about the effectiveness of ITNs was not high. In addition, from 2014 indoor residual spraying was implemented in the area [34], and as a result it is possible that the intense campaign shifted attention away from ITNs. Several studies on ITN use in different countries have indicated that a lack of value attached to ITNs (or low response efficacy) plays a large role in their use, citing that without proper information, populations are not able to connect malaria prevention with ITNs [23, 35, 36]. These studies contend that it is necessary for malaria prevention programs to highlight that ITNs not only prevent malaria, but also provide other direct benefits such as averting financial costs through reduced hospital visits.

Another reason for not finding any association between response efficacy and ITN use could be attributed to how the construct was measured. Further work to improve the construct validity of the response efficacy sub-scale is recommended.

With the EPPM, Witte asserted that the extent to which an individual feels threatened by a health risk determines his or her motivation to act, but confidence to effectively reduce or prevent the threat determines the action itself [14]. In this case, with threat levels and self-efficacy being high, there is a need to increase the response efficacy perceptions among the population, and perhaps reduce perceived severity (fear).

This study had some limitations. The internal consistency used to measure some of the EPPM constructs was poor to acceptable, suggesting that further work on the measurement applied to ITN use needs to be carried out. As a result of poor internal consistency for the perceived severity scale, a single item was used. We looked at how the individual item performed, and the results were similar to the overall scale. Because ITN use was self-reported, it is possible that bednet use the previous night was over-reported.

Another limitation is the potential to generalize the results beyond the study site. Hence, it is difficult to conclude that the findings are applicable to pregnant women in other parts of Uganda. The response efficacy scale had low internal consistency and factor analysis showed that items designed to measure it loaded on other sub-scales. This suggests that more work is needed on the response efficacy items used in the *Malaria Behavior Change Communication (BCC) Indicator Reference Guide* across different endemic settings. In addition, as pregnant women were recruited from antenatal care we were unable to interview their husbands/partners who may be critical in the decision-making related to Net use.

## Conclusions

Self-efficacy was the most important predictor of ITN use. Programs targeting ITN use among pregnant women should seek to strengthen perceived self-efficacy to achieve improved use. High levels of perceived severity can elicit a fear response which is counterproductive. In this study high threat was not linked to ITN use although it was associated with consistent Net use, suggesting that there was an effective danger response. EPPM provides a useful framework for analysing and understanding audiences' threat and efficacy perceptions and their association with behavioural outcomes. The study found that the threat and self-efficacy constructs in the model explained consistent ITN use among pregnant women. These findings can guide future interventions to maintain ITN use in malaria endemic settings. Specifically, interventions need to address messages and other interventions to increase self-efficacy to use ITNs together with sustaining high perceptions of the threat of malaria.

## Supporting information

**S1 Dataset.**
(XLSX)

**S1 Fig.**
(TIF)

**S1 Questionnaire.**
(DOCX)

## Acknowledgments

The authors wish to thank the Tororo district general hospital authorities for their support in the data collection and operationalisation of the study and the respondents of the survey questionnaires.

## Author Contributions

**Conceptualization:** Charles Nelson Kakaire.

**Supervision:** Nicola Christofides.

**Writing – original draft:** Charles Nelson Kakaire.

**Writing – review & editing:** Charles Nelson Kakaire, Nicola Christofides.

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
