## [Decision Letter · Decision Letter 0]

9 Aug 2022

PONE-D-22-13933The role of perceived threat and self-efficacy in the use of Insecticide Treated Bednets (ITNs) to prevent malaria among pregnant women in Tororo District, Uganda

PLOS ONE

Dear Dr.  Kakaire,

Thank you for submitting your manuscript to PLoS ONE. After careful consideration, we felt that your manuscript requires substantial revision, following which it can possibly be reconsidered, thus governing the decision of a “major revision”. As requested by the reviewers, the authors need to address several concerns, particularly related to the methods, including questionnaire. For your guidance, a copy of the reviewers' comments was included below.

Please submit your revised manuscript by Sep 23 2022 11:59PM

. If you will need more time than this to complete your revisions, please reply to this message or contact the journal office at plosone@plos.org. Please include the following items when submitting your revised manuscript:A rebuttal letter that responds to each point raised by the academic editor and reviewer(s). You should upload this letter as a separate file labeled 'Response to Reviewers'.A marked-up copy of your manuscript that highlights changes made to the original version. You should upload this as a separate file labeled 'Revised Manuscript with Track Changes'.An unmarked version of your revised paper without tracked changes. You should upload this as a separate file labeled 'Manuscript'.

We look forward to receiving your revised manuscript.

Kind regards,

Luzia H Carvalho, Ph.D.

Academic Editor

PLOS ONE

Journal Requirements:

   "NO - The funders had no role in study design, data collection and analysis, decision to publish, or preparation of the manuscript."

4.Thank you for stating the following in the Acknowledgments Section of your manuscript: 

   "The authors wish to thank the Tororo district general hospital authorities for their support in the data collection and operationalisation of the study and the respondents of the survey questionnaires. This work was partly funded by the Malaria Consortium Uganda. The contents do not necessarily reflect the views of Malaria Consortium Uganda. "

  "NO - The funders had no role in study design, data collection and analysis, decision to publish, or preparation of the manuscript."

6. We note you have included a table to which you do not refer in the text of your manuscript. Please ensure that you refer to Table 2,4 and 5 in your text; if accepted, production will need this reference to link the reader to the Table.

Reviewers' comments:

Reviewer's Responses to Questions

**Comments to the Author**

1. Is the manuscript technically sound, and do the data support the conclusions?

Reviewer #1: Yes

Reviewer #2: Partly

2. Has the statistical analysis been performed appropriately and rigorously? 

Reviewer #1: I Don't Know

Reviewer #2: Yes

3. Have the authors made all data underlying the findings in their manuscript fully available?

Reviewer #1: Yes

Reviewer #2: Yes

4. Is the manuscript presented in an intelligible fashion and written in standard English?

Reviewer #1: Yes

Reviewer #2: No

5. Review Comments to the Author

Reviewer #1: Manuscript Title: The role of perceived threat and self-efficacy in the use of Insecticide Treated Bednets (ITNs) to prevent malaria among pregnant women in Tororo District, Uganda

Manuscript ID: PONE-D-22-13933

Reviewer comments

General comments: This is an important and unique study on ITN use among pregnant women looking the important role played by perceived susceptibility, severity, self-efficacy and response efficacy. However, this paper needs some major revisions before being considered for publication.

Specific comments

Introduction: “While the Extended Parallel Process Model has been tested in several interventions including workplace safety [13], asthma intervention programs [14], and physicians’ decision to test patients for kidney disease [15] among others, there is no known study that has tested the application of the model in a malaria setting, specifically ITN use among pregnant women”.

#1 This is a very important statement as it highlights the contribution this study will add to the body of knowledge. Could the authors add to this statement by highlighting the importance of the four EPPM construct in malaria prevention (i.e ITN use) and how this had impacted other health interventions and behaviours

Methods-Questionnaire

#2 I am wondering why the authors measured the four EPPM construct using 4-point scale response instead of the known 5 point likert scale response.

#3 Authors should give details on how the 4-point scale responses were recategorized into high, moderate and low

#4 I am wondering why the authors considered only age, marital status, educational attainment and socio-economic status as the covariates. Were they guided by literature?

I thought a decision of a pregnant woman to sleep under ITN is also highly influenced by their partners (husband) characteristics. It could be interesting to control for the effect of the husband characteristics on EPPM construct and pregnant women ITN use relationship since the study setting (Uganda) is where African’s men are major decision makers.

#5 Can the authors kindly provide details on how “consistency of ITN use” was measured?

What constitutes consistency?

Ethics

“Ethical approval was obtained from the University of Witwatersrand Faculty of Health

Sciences Human Research Ethics Committee, in South Africa, the School of Health Sciences

Research and Ethics committee at Makerere University in Uganda and the Uganda National

Council for Science and Technology”.

#5 Could authors kindly provided the ethical protocol number?

Results

# 6 In Table 4 and 5. I am wondering whether the authors used row or column percentage in calculating the association between socio-demographic characteristics and ITN use. This is because I am struggling to see how the percentage were obtained. Authors should kindly clarify

# 7 In Table 5, Authors considered “inconsistent ITN use and no ITN use as one” Please clarify or provide some explanation to why a pregnant woman who is inconsistently using ITN will be considered as the same with another who is not using ITN.

Conclusion

“We found that the threat and self-efficacy constructs in the model explained consistent ITN use among pregnant women. These findings can guide future interventions to maintain ITN use in malaria endemic settings”.

#8. Authors should give a detailed explanation on how EPPM constructs can guide future ITN usage in malaria endemic areas.

Reviewer #2: Thank editor and authors for giving me a chance to read this manuscript. The study pointed out a public health problem i.e. ITN use among pregnant women. Overall, the manuscript is well written but for me, I think the methods and discussion need major attention to publish it at the PlosOne.

6. PLOS authors have the option to publish the peer review history of their article (what does this mean?). If published, this will include your full peer review and any attached files.

Reviewer #1: No

Reviewer #2: No

---

## [Author Response · Author response to Decision Letter 0]

19 Dec 2022

See attached Response to Reviewer comments file

---

## [Decision Letter · Decision Letter 1]

1 Feb 2023

PONE-D-22-13933R1The role of perceived threat and self-efficacy in the use of Insecticide Treated Bednets (ITNs) to prevent malaria among pregnant women in Tororo District, UgandaPLOS ONE

Dear Dr. Kakaire,

Thank you for resubmitting your manuscript to PLoS ONE. Although the data from this study has potential to be informative, relevant topics raised by the reviewer #2 during the peer review process remain to be addressed by the authors. Unfortunately, the authors did not submit a rebuttal letter that responds to each point raised by the reviewers as required by the publication policy of PLoS journals. At this time, we strongly suggest the authors to proper address all topics raised by the reviewers.  For your guidance, a copy of the reviewer’s comments was included below.   Please submit your revised manuscript by Mar 18 2023 11:59PM. If you will need more time than this to complete your revisions, please reply to this message or contact the journal office at plosone@plos.org. Please include the following items when submitting your revised manuscript:A rebuttal letter that responds to each point raised by the academic editor and reviewer(s). You should upload this letter as a separate file labeled 'Response to Reviewers'.A marked-up copy of your manuscript that highlights changes made to the original version. You should upload this as a separate file labeled 'Revised Manuscript with Track Changes'.An unmarked version of your revised paper without tracked changes. You should upload this as a separate file labeled 'Manuscript'.

We look forward to receiving your revised manuscript.

Kind regards,

Luzia H Carvalho, Ph.D.

Academic Editor

PLOS ONE

Reviewers' comments:

Reviewer's Responses to Questions

**Comments to the Author**

1. If the authors have adequately addressed your comments raised in a previous round of review and you feel that this manuscript is now acceptable for publication, you may indicate that here to bypass the “Comments to the Author” section, enter your conflict of interest statement in the “Confidential to Editor” section, and submit your "Accept" recommendation.

Reviewer #1: All comments have been addressed

Reviewer #2: (No Response)

2. Is the manuscript technically sound, and do the data support the conclusions?

Reviewer #1: Yes

Reviewer #2: Partly

3. Has the statistical analysis been performed appropriately and rigorously? 

Reviewer #1: Yes

Reviewer #2: Yes

4. Have the authors made all data underlying the findings in their manuscript fully available?

Reviewer #1: Yes

Reviewer #2: Yes

5. Is the manuscript presented in an intelligible fashion and written in standard English?

Reviewer #1: Yes

Reviewer #2: No

6. Review Comments to the Author

Reviewer #1: The authors should include their inability to include partners' characteristics as a limitation of the study

Reviewer #2: Thank authors for addressing my comments in previous round of review. However, I could not find the intensive revisions upon some of my comments. I have highlighted these points where authors still need to address. Moreover, I put some more comments in red color for more clarifications.

7. PLOS authors have the option to publish the peer review history of their article (what does this mean?). If published, this will include your full peer review and any attached files.

Reviewer #1: No

Reviewer #2: No

---

## [Author Response · Author response to Decision Letter 1]

19 Apr 2023

Dear Editor, 

We thank the reviewers for their careful and attentive comments on our manuscript. Below, we respond point-by-point to the comments and suggestions and demonstrate changes made to the paper.

Sincere thanks,

Abstract

- Use standard keywords (e.g., mesh-engine) (I did not see any revision)

Thank you for flagging the need to ensure that keywords match mesh-engine. This has been updated to the standard keywords e.g. insecticide treated bed nets, risk reduction behaviors. 

- Word usage should be consistent (e.g., AOR or aOR?) 

This has been updated

- What does it mean the communication interventions? Use the specific word. 

Thank you for pointing out the need to clarify the term used. This has been updated to social and behavior change communication interventions

Introduction

- Expand/ Include absolute number for magnitude of malaria problem in Uganda. (Not only mentioned the malaria problem is high, please add the real number of malaria problem e.g we can reference to WHO world malaria report 2022) 

This paragraph has been updated to include numbers; incidence rates at above 154 cases per 1,000 people

- WHO: only use long form in first appearance.

 This has been updated.

- Needs description about malaria problem among pregnant women. How were previous study findings? (I did not see author response on it) 

Thanks for pointing out the gap. We added the following: Several poor health outcomes are associated with malaria in pregnancy including stillbirth, preterm birth, maternal and neonatal mortality, congenital malaria, maternal anaemia and low birth weight [4] A study by Namusoke and others noted that malaria infection rates among pregnant women were 15.5% (59/380) of active infections and 4.5% (17/380) of past infections [5].

Methods

- Needs detailed inclusion and exclusion criteria for sampling the pregnant women (how was the exclusion criteria for example pregnant women migrated from another area)

Thank you for the comment. We have added sentence. 

- Includes detailed sample size calculation

A detailed description of sample size calculation has been added

- Given that there were a lot of IRB committees there, put IRB-number of each approval. 

This has been updated, with the respective IRB numbers included as follows; University of Witwatersrand Faculty of Health Sciences Human Research Ethics Committee, in South Africa (#M140860), the School of Health Sciences Research and Ethics committee at Makerere University in Uganda (#256) and the Uganda National Council for Science and Technology (# SS 3656).

- Are there any criteria for independent variables to be included in logistics regression models?

We selected variables that were statistically significant or approaching significance

- Consisting of a map locating study area shall be considered for the readers know the

geographical situation. 

This has been updated and the map included as Figure 2

Results

- Why did the samples different for educational attainment? How did you control it during analysis? 

Educational attainment data was missing for 28 participants. We controlled for educational attainment in the analysis. But we didn’t impute for missing data.

- How did you get sample size 46 for asking reasons for non-ITN use? 

Of the 49 participants who reported not using an ITN the previous night, 46 provided a reason. We added a sentence to explain this in the results.

- Usually, we didn’t use the word Associations for the results out from Chi-square only. (Your table 4 and 5) (should correct in the paragraph as well) 

Thank you for pointing this out. We have made changes to the paragraphs as well.

- Table 5: How did you categorize Low, moderate, and high groups for the independent

variables? Are there any theories or references? (Please also add a relevant citation for the criteria use e.g., bloom classification) 

We used tertiles to categorise a continuous variable into the three groups; Low moderate and high.

- For logistics regression, better to include (n) for each group rather than putting only AOR and 95% CIs. 

The “n” is included in the table title

Discussion

- Some sentences are misleading (e.g., Page 18, Line 306-308 I believed current study did not research on insecticide resistance). (The literatures might be relevant with the title, but we usually discussed what we found from the study findings/ results. I still believed current study did not research on insecticide resistance)

We removed the sentences on insecticide resistance as we agree that the text is better without it.

Other comments

- Language should be improved throughout the manuscript. (I still see many errors throughout the manuscript) 

This has been updated.

- Although the study was conducted in 2015, better to use the data from recent reports or papers. Almost all references are a bit old and need to be changed with recent ones.

We have added some more recent references.

- Also, references styles need to be rechecked and revised. (e.g., journal names should follow PubMed acronyms). Check against Plos in-house style. (I did not see author corrections on the reference styles. Most of the reference are wrong especially authors names, year of publication and the journal names)

We have checked and edited the references

- Include study tools as supplementary file(s). 

This has been added as S2.

---

## [Decision Letter · Decision Letter 2]

22 May 2023

PONE-D-22-13933R2The role of perceived threat and self-efficacy in the use of Insecticide Treated Bednets (ITNs) to prevent malaria among pregnant women in Tororo District, UgandaPLOS ONE

Dear Dr. Kakaire,

Thank you for submitting your manuscript to PLoS ONE. After careful consideration, we feel that your manuscript will likely be suitable for publication if the authors revise it to address specific points raised now by the reviewer. According to the reviewer, there are some specific areas where further improvements would be of substantial benefit to the readers.   For your guidance, a copy of the reviewers' comments was included below.

We look forward to receiving your revised manuscript.

Kind regards,

Luzia H Carvalho, Ph.D.

Academic Editor

PLOS ONE

Journal Requirements:

Reviewers' comments:

Reviewer's Responses to Questions

**Comments to the Author**

1. If the authors have adequately addressed your comments raised in a previous round of review and you feel that this manuscript is now acceptable for publication, you may indicate that here to bypass the “Comments to the Author” section, enter your conflict of interest statement in the “Confidential to Editor” section, and submit your "Accept" recommendation.

Reviewer #2: All comments have been addressed

2. Is the manuscript technically sound, and do the data support the conclusions?

Reviewer #2: Yes

3. Has the statistical analysis been performed appropriately and rigorously? 

Reviewer #2: Yes

4. Have the authors made all data underlying the findings in their manuscript fully available?

Reviewer #2: No

5. Is the manuscript presented in an intelligible fashion and written in standard English?

Reviewer #2: Yes

6. Review Comments to the Author

Reviewer #2: Dear Authors;

I acknowledge the improvements made, but I believe that the manuscript could benefit further from some minor revisions.

1. One area that requires attention is the outdated references. For instance, the World Malaria Report 2022 is now available, yet the authors have cited the WMR 2014. It is crucial to incorporate the latest and most relevant references to ensure the manuscript's accuracy and currency.

2. Another aspect that needs addressing is the reference style. It is important to adhere to standard conventions. For example, using acronyms for journals is a common practice. As an illustration, "Malaria Journal" is typically abbreviated as "Malar J." I recommend consulting the NLM catalog for further guidance.

3. I suggest considering the use of appropriate keywords for enhanced visibility. Utilizing the MESH Engine at https://www.ncbi.nlm.nih.gov/mesh/ can aid in identifying relevant keywords that will improve the discoverability and reach of the manuscript.

4. In the discussion section, it would be beneficial to include insights from the authors regarding the findings of the current study based on the geographical context of the study area and explore possible reasons for the observed results. By doing so, the discussion can provide a more comprehensive understanding of the study's implications.

By addressing these concerns, I believe the manuscript can be further improved, ensuring its relevance and accessibility to a wider audience.

7. PLOS authors have the option to publish the peer review history of their article (what does this mean?). If published, this will include your full peer review and any attached files.

Reviewer #2: No

---

## [Author Response · Author response to Decision Letter 2]

4 Jul 2023

A rebuttal letter has been attached with individual responses to each of the comments raised by the two Reviewers.

---

## [Decision Letter · Decision Letter 3]

12 Jul 2023

The role of perceived threat and self-efficacy in the use of Insecticide Treated Bednets (ITNs) to prevent malaria among pregnant women in Tororo District, Uganda

PONE-D-22-13933R3

Dear Dr. Kakaire,

We’re pleased to inform you that your manuscript has been judged scientifically suitable for publication and will be formally accepted for publication once it meets all outstanding technical requirements.

Kind regards,

Luzia H Carvalho, Ph.D.

Academic Editor

PLOS ONE

Additional Editor Comments (optional):

Reviewers' comments:

Reviewer's Responses to Questions

**Comments to the Author**

1. If the authors have adequately addressed your comments raised in a previous round of review and you feel that this manuscript is now acceptable for publication, you may indicate that here to bypass the “Comments to the Author” section, enter your conflict of interest statement in the “Confidential to Editor” section, and submit your "Accept" recommendation.

Reviewer #2: All comments have been addressed

2. Is the manuscript technically sound, and do the data support the conclusions?

Reviewer #2: Yes

3. Has the statistical analysis been performed appropriately and rigorously? 

Reviewer #2: Yes

4. Have the authors made all data underlying the findings in their manuscript fully available?

Reviewer #2: Yes

5. Is the manuscript presented in an intelligible fashion and written in standard English?

Reviewer #2: Yes

6. Review Comments to the Author

Reviewer #2: The authors have now addressed almost all of my comments, and I find the manuscript is acceptable now. However, there are still some remaining grammatical errors throughout the manuscript and formating errors in the references. I hope the authors can improve these issues during the proofreading process.

7. PLOS authors have the option to publish the peer review history of their article (what does this mean?). If published, this will include your full peer review and any attached files.

Reviewer #2: No

---

## [Editor Report · Acceptance letter]

17 Jul 2023

PONE-D-22-13933R3 

The role of perceived threat and self-efficacy in the use of Insecticide Treated Bednets (ITNs) to prevent malaria among pregnant women in Tororo District, Uganda 

Dear Dr. Kakaire:

I'm pleased to inform you that your manuscript has been deemed suitable for publication in PLOS ONE. Congratulations! Your manuscript is now with our production department. 

Kind regards, 

on behalf of

Dr. Luzia H Carvalho 

Academic Editor

PLOS ONE